# Pathogenic and Genetic Diversity of *Sclerotium rolfsii,* the Causal Agent of Southern Blight of Common Bean in Uganda

**DOI:** 10.3390/jof12010018

**Published:** 2025-12-26

**Authors:** Samuel Erima, Moses Nyine, Mildred Ochwo Ssemakula, Geoffrey Tusiime, Eduard Akhunov, Alina Akhunova, Ural Yunusbaev, Emmanuel Amponsah Adjei, Settumba B. Mukasa, Michael Hilary Otim, Thomas Lapaka Odong, Allan Nkuboye, Agnes Candiru, Pamela Paparu

**Affiliations:** 1National Crops Resources Research Institute, National Agricultural Research Organization, Namulonge, Kampala P.O. Box 7084, Uganda; motim9405@gmail.com (M.H.O.); nkuboyeallan2019@gmail.com (A.N.); candiruagnes95@gmail.com (A.C.); pamela.paparu@gmail.com (P.P.); 2Department of Crop Science and Horticulture, School of Agricultural Sciences, Makerere University, Kampala P.O. Box 7062, Uganda; mildred.ochwossemakula@mak.ac.ug (M.O.S.); gwtusiime@gmail.com (G.T.); sbmukasa@gmail.com (S.B.M.); thomas.i.odong@gmail.com (T.L.O.); 3Department of Crop Science, Faculty of Agriculture, Muni University, Arua P.O. Box 725, Uganda; 4International Institute of Tropical Agriculture, PMB 5320, Oyo Road, Ibadan 200001, Oyo State, Nigeria; m.nyine@cgiar.org; 5Wheat Genetics Resources Center, Kansas State University, Manhattan, KS 66506, USA; eakhunov@ksu.edu (E.A.); ural@ksu.edu (U.Y.); 6Department of Plant Pathology, Kansas State University, Manhattan, KS 66506, USA; akhunova@ksu.edu; 7Integrated Genomics Facility, Kansas State University, Manhattan, KS 66506, USA; 8Department of Crop Improvement, CSIR-Savanna Agriculture Research Institute, Tamale P.O. Box TL 52, Ghana; emmaadjei1@gmail.com

**Keywords:** pathogenicity, genetic diversity, characterisation, southern blight, common bean

## Abstract

*Sclerotium rolfsii* Sacc. is a soil-borne fungus that causes southern blight on many crops in the tropical and subtropical regions. In 2018, southern blight was reported as the most prevalent bean root rot in Uganda. Earlier studies ascertained the morphological and pathogenic diversity of *S. rolfsii*, but a limited understanding of its genetic diversity exists. Knowledge of *S. rolfsii* genetic diversity is a critical resource for pathogen surveillance and developing common bean varieties with durable resistance. A total of 188 *S. rolfsii* strains from infected common bean plants were collected from seven agro-ecological zones of Uganda in 2013, 2020 and 2021, and characterized morphologically and pathogenically. The genetic diversity of the strains was assessed using single-nucleotide polymorphisms (SNPs) obtained from whole-genome sequencing. The growth rate of the strains ranged between 1.1 and 3.6 cm per day, while the number of sclerotia produced ranged from 0 to 543 per strain. The strains had fluffy, fibrous, and compact colony texture. The strains were pathogenic on common bean and caused disease severity indices ranging from 10.1% to 93.3%. Average polymorphic information content across all chromosomes was 0.27. Population structure analysis identified five genetically distinct clusters. The results of analysis of molecular variance revealed that 54% of the variation was between clusters while 46% of variation was within clusters. Pairwise comparison of Wright’s fixation indices between genetic clusters ranged from 0.31 to 0.78. The findings of this study revealed moderate genetic diversity among *S. rolfsii* strains, which should be taken into consideration when selecting strains for germplasm screening.

## 1. Introduction

The common bean (*Phaseolus vulgaris* L.) is the third most important legume in the world after soybeans (*Glycine max* L.) and peanuts (*Arachis hypogaea* [1,2]. In Uganda, it is the third most important annual crop after maize (*Zea mays* L.) and cassava (*Manihot esculenta* C) [3]. In 2019, common bean production declined to 438,000 tons from 1,008,410 tons in 2016, yet the area planted had increased to 800,000 hectares from 670,737 hectares, with an average yield of 0.6 tons per hectare [3,4]. Uganda Bureau of Statistics (UBOS) attributed this decline to reduced productivity. Factors known to negatively influence common bean productivity include the following: poor soil fertility, pests (bean stem maggot, aphids, bean leaf beetles, etc.) and diseases (root rots, angular leaf spot, and anthracnose, etc.), poor agronomic practices, weed competition, moisture stress, and lack of quality seed [5].

*Sclerotium rolfsii* Sacc. is a soil-borne pathogen that primarily attacks the root and stem but can also attack the pods, petioles, and leaves when these parts come in contact with soil [6]. The species was first described in 1911 by Italian mycologistAndrea Saccardo, but it was not until 1982 that it was first identified as the causal agent of Southern blight in tomatoes (*Solanum lycopersicum* L.) [7]. *Sclerotium rolfsii* is the anamorphic stage (asexual state) of the pathogen, while *Athelia rolfsii* is the teleomorph state (sexual state) and is rarely observed [8]. The pathogen is classified as follows: Kingdom—Fungi, Phylum—Basidiomycota, Subphylum—Agaricomycotina, Class—Agaricomycetes, Order—Atheliales, Family—*Atheliaceae*, and Genus—*Athelia*/*Sclerotium* [6]. *Sclerotium rolfsii* is commonly found in the tropics, subtropics, and warm temperate regions, especially in the United States, Central and South America, West Indies, Southern Europe, India, Japan, the Philippines, and Hawaii. The pathogen rarely occurs when temperatures drop below 0 °C [6].

Paparu et al. [9] reported southern blight as the most prevalent bean root rot in Uganda, besides Fusarium, Pythium, and Rhizoctonia root rot diseases. The pathogen causes both pre- and post-emergence dumping-off, with the former mostly observed in highly susceptible varieties and for highly virulent strains. For post-emergence dumping-off, wilting can occur as early as one week after planting [10]. There is general wilting of the plant characterized by dry shoot and plant death. Under conditions of high humidity, abundant white mycelia are usually visible on the collar and root of infected plants. Brown sclerotia are usually visible at the base of the plant at advanced stages of wilting [11]. Yield losses due to southern blight in the field range from 55.5 to 64.9% [12], yet in Uganda, there is no common bean variety known to be resistant to the pathogen [9]. Use of host plant resistance is the most cost-effective and sustainable strategy for the management of root rots by the smallholder farmers who are the major bean growers in sub-Saharan Africa [13,14]. The knowledge of pathogen diversity is key in efforts to develop durable resistance in crops.

Pathogen diversity can be determined using morphological/cultural, pathogenic, and genetic studies. Morphological/cultural diversity studies are often cheap, but they may not distinguish strains that produce similar morphological structures, and they require experienced mycologists to conduct them [15]. Pathogenic diversity studies aid in determination of the host range of the pathogen and the selection of strains which can be used for screening the germplasm [16]. Genetic diversity of *S. rolfsii* has been studied using PCR-based molecular markers including the Internal Transcribed Spacers (ITS) region of the ribosomal RNA [17], Translation Elongation Factor 1 alpha gene region (TEF 1 α) [18], Randomly Amplified Polymorphic DNA (RAPD) [19] and Amplified Fragment Length Polymorphism (AFLP) [20]. However, these amplicon sequencing (AmpliSeq) approaches require up front variant discovery to design and synthesize the primer sets [21]. In the last decade, next generation sequencing (NGS) has been utilized [22,23], with approaches such as genotyping by sequencing (GBS) and restriction site-associated DNA sequencing (RAD Seq) used for diversity studies. Though the RAD Seq and GBS do not require prior sequence information, they are characterized by reduced representation of the genome [24]. Due to advances in the NGS technologies and reduction in the cost of sequencing, skim sequencing (SkimSeq) of the whole genome at low coverage can be used to identify a large number of polymorphic markers cost-effectively [21,24]. Availability of abundant single nucleotide polymorphism (SNP) markers facilitates diversity studies in organisms [25]. However, to apply next generation sequencing, high-quality DNA is needed. Obtaining high-quality DNA from *S. rolfsii* is reported to be challenging due to the presence of extracellular polysaccharides in their cell walls [26]. However, some modifications in the DNA extraction methods are reported to effectively remove these polysaccharides [26,27,28].

Although Paparu et al. [10] studied the morphological and pathogenic diversity of *S. rolfsii* strains collected before 2020, the genetic diversity of these strains was never characterized. Therefore, this study intended to (1) determine the morphological diversity of strains not previously studied, (2) determine the pathogenicity of all *S. rolfsii* strains following long periods of storage, and (3) conduct genetic diversity and population structure analyses of the *S. rolfsii* strains using whole-genome re-sequencing data. The genetic differentiation among the sub-populations of *S. rolfsii* was compared with the agro-ecology of strains’ origins, growth rate, and number of sclerotia produced to assess their potential role in pathogen adaptation and virulence on the common bean. The generation of this information can facilitate pathogen surveillance and selection of strains to screen germplasm for breeding disease-resistant varieties.

## 2. Materials and Methods

### 2.1. Morphological Diversity of Selected Sclerotium rolfsii Strains

Eighty-four (84) *S. rolfsii* strains collected in the years 2020 and 2021 and not previously characterized morphologically were used in this study. The pathogen was isolated according to the procedure by Paparu et al. [10]. Morphological diversity was studied by assessing the growth rate, colony texture, and the number of Sclerotia produced by each strain. Strains that had been stored on filter papers were initiated on PDA (Esvee Biologicals, Mumbai, India). The strains were purified by hyphal tipping according to the procedure by Paparu et al. [10]. We used potato dextrose agar (PDA) with 0.3 g of rifamycin per liter to conduct morphological characterization according to the procedure by Paparu et al. [10]. Inoculation was performed by touching the surface of the fungal culture on a Petri dish with a needle and then transferring the attached mycelia to the center of the Petri dish. Each fungal strain was inoculated in triplicate onto 8cm diameter Petri dishes and incubated in the dark at 25 °C. Colony diameter was measured daily, beginning 2 days post-inoculation, and continued until the mycelium fully covered the plate surface, which occurred by day 4 for most strains. Following growth assessment, colony color and texture were documented. Cultures were maintained for 28 days, after which the number of sclerotia was recorded following the protocol described by Mullen [29].

### 2.2. Pathogenic Diversity of Sclerotium rolfsii Strains

#### 2.2.1. *Sclerotium rolfsii* Strains

The pathogenicity of 188 *S. rolfsii* strains, including 104 strains previously studied by Paparu et al. [10], and the 84 strains collected in 2020 and 2021 was investigated in this study. Pure cultures of these strains preserved on filter paper and stored in Eppendorf tubes at 4 °C were tested for pathogenicity for the first time following a long period in storage. The strains were collected from several agro-ecological zones including the Eastern Highlands (EH), Lake Victoria Crescent and Mbale farmlands (LVC), Northern Mixed Farming System (NMFS), South-Western Highlands (SWH), Teso Farming Zone (TFZ), Western Mixed Farming System (WMFS), and West Nile Mixed Farming System (WNFS). Three strains out of the188 were from Bukoba in Tanzania.

#### 2.2.2. Preparation of *Sclerotium rolfsii* Inoculum

*Sclerotium rolfsii* pathogenicity was conducted according to procedure by by Paparu et al. [10]. A total of 188 strains were screened against five common bean lines: MLB49-89A (Fusarium root rot-resistant check), RWR719 (Pythium root rot-resistant check), KWP9 (southern blight-tolerant check), NABE 14 (released variety, tolerant to Fusarium root rot and CAL96, also known as K132 (susceptibility check for common bean diseases)) [30]. To prepare the inoculum, 50 g of millet was mixed with 50 mL of water in an autoclave bag and sterilized at 121 °C for 1 h. Five 1 cm^2^ agar plugs were excised from 14-day-old PDA cultures (39 g/L) grown in 8 cm Petri dishes. The plugs were aseptically mixed with sterilized millet in a laminar flow cabinet, sealed in bags, and incubated at 25 °C for 21 days.

#### 2.2.3. Soil Inoculation and Planting of Common Bean Lines

Twenty kilograms (20 kg) of steam-sterilized loam soil and sand mixed in a 2:1 ratio were placed in the 70 cm × 35 cm × 10 cm wooden trays lined with a polythene sheet after disinfecting with 75% ethanol. The soil was then inoculated with 10 g of millet carrying fungal mycelia and sclerotia. Each strain was inoculated in a single tray and replicated once. Trays were arranged in a Completely Randomized Design (CRD) in the screen house. In each tray, 16 seeds of each line were planted in two rows. Common bean lines were randomized in each tray in a split plot design. Sixteen seeds of each of the test lines planted in an un-inoculated tray were used as controls. The planting trays were protected from direct rainfall but watered daily until the experiments were terminated. Strains were evaluated in five separate experiments (each repeated once), with 58 strains screened in experiment 1, 45 in experiment 2, 26 in experiment 3, 28 in experiment 4, and 31 in experiment 5.

#### 2.2.4. Assessment of Seed Germination, Southern Blight Incidence, and Severity

Fourteen days after planting, germination data was collected. The seeds that did not germinate were examined to assess pre-emergence dumping-off (characterized by rotting and fungal growth on seed) and non-viability (intact seed with no sign of rotting or fungal growth). Plants that germinated were assigned germination score 1, while those that did not germinate were assigned score 0. Preliminary data on incidence and severity was also collected at this point, where plants that died of pre-emergence dumping-off were assigned score 1 for incidence and severity score 6 for pre-emergence dumping-off. Final southern blight severity was assessed 28 days after planting using a scale developed by Le et al. [16] and modified by Paparu et al. [10] by adding a score 6 to represent pre-emergence dumping-off (Figure 1).

### 2.3. Genetic Diversity of Sclerotium rolfsii Strains

*Sclerotium rolfsii* produces extracellular polysaccharides which co-precipitate with DNA, adversely affecting DNA recovery [31]. Male et al. [26] also reported that DNA extraction kits do not work well for *S. rolfsii*. Hence, a modification of the protocol by Joint Research Centre (JRC) European Commission [32] was adopted, where dry mycelia was used and precipitated on ice using absolute isopropanol and 2.5 M sodium acetate.

A DNA extraction buffer (2 M NaCl, 0.2 M EDTA, 0.2 M Tris HCL (PH8), and 1% SDS) was prepared and autoclaved. Mycelia from two-week-old culture were scrapped into sterile Petri dishes and oven-dried overnight at 35 °C. About 0.02 g of the dry mycelia was put in 2 mL Eppendorf tubes containing beads and ground at 1400 revolutions per minute for 3 min in a Geno grinder (1600 MiniG, Cole-Parmer, Chicago, IL, USA). Then 800 µL of DNA extraction buffer preheated to 65 °C was added to the ground mycelia and ground for another 2 min in the Geno grinder (1600 MiniG, Cole-Parmer, Chicago, USA). The tubes were then incubated in a water bath at 65 °C for one hour, followed by centrifugation at 12,000 revolutions per minute (rpm) for 10 min. The supernatant (500 µL) was transferred into a sterile 2 mL Eppendorf tube. An equal volume of chloroform and isoamyl alcohol (24:1) was added, mixed for 2 min, and centrifuged at 12,000 rpm for 10 min to separate the DNA from the cellular residues. If a clear supernatant was not obtained, this step was repeated. The supernatant (400 µL) was transferred into a 1.5 mL sterile Eppendorf tube, mixed with an equal volume of ice cold isopropanol stored at −20 °C by inverting a tube 5 to 10 times. Then, 40 µL of 2.5 M sodium acetate was added to each tube and inverted 10 times.

The solution was incubated at −20 °C for two hours to precipitate DNA. Samples were centrifuged at 12,000 rpm for 10 min to pellet the DNA, and the supernatant was discarded. The pellets were washed twice with 500 µL of 75% ethanol and left to air dry at room temperature for 1 h. After this, the DNA was dissolved in 100 µL of 1× TE buffer (10 mM Tris HCL (pH 8), 1 mM EDTA). DNA was quantified using a NanoDrop (ND-1000) (Thermo Fisher, Waltham, MA, USA). The integrity and quality of DNA was assessed using electrophoresis with 1% *w*/*v* agarose gel (Appendix A). The extracted DNA was shipped to Kansas State University, where further assessment of the DNA quality and integrity was performed using a TapeStation 4200 (Avantor, Radnor, PA, USA).

Libraries were prepared using Nextera DNA library preparation protocol (Illumina, San Diego, CA, USA) (Nextera DNA Library Prep Protocol Guide (1000000006836) (https://support.illumina.com/content/dam/illumina-support/documents/documentation/chemistry_documentation/samplepreps_nextera/nexteradna/nextera-dna-library-prep-protocol-guide-1000000006836-00.pdf, accessed on 23 April 2024)) with reduced reaction volume) [21,33]. The libraries were subjected to size selection using the Pippin Prep system (Sage Scientific, Beverly, MA, USA) to enrich for 400–600 bp fragments. Barcoded libraries were pooled and sequenced on NovaSeq 6000 (San Diego, CA, USA) to generate ~16.5× coverage of 2 × 150 bp paired-end reads per sample.

### 2.4. Data Analysis

#### 2.4.1. Analysis of Germination, Incidence, and Severity Data

Percentage germination for the 16 plants of each common bean variety was calculated in Microsoft Excel. The severity of southern blight was calculated according to the formula by Chiang et al. [34].



$$DSI=\frac{\sum (Class\;frequency\times Score\;rating\;of\;class)}{\left(Total\;number\;of\;observation\right)\times \left(Maximum\;disease\;index\right)}\times 100$$



Generalized linear mixed model (GLMM) was used to analyze the disease severity data, where the tray number was fitted in the model as a random effect. Where there were significant difference in the DSI, pairwise comparison of *p* values was conducted to determine which variables differed. The Pearson’s Chi-square test (X^2^) was used to test for association between pathogenicity and geographical origin of the strains. Correlation analyses were conducted between growth rate, pathogenicity, and number of sclerotia produced.

#### 2.4.2. Analysis of *Sclerotium rolfsii* Sequence Data

Sequence reads were demultiplexed using a custom python script to separate individual sample reads based on the index barcodes. Adaptor trimming and read quality filtering was performed using Trimmomatic v0.32 software [35]. The high-quality reads were aligned to the indexed reference genome of *Agroathelia rolfsii* ASM1834391v1 by Yan et al. [22] using Burrows-Wheeler Alignment Tool (bwa-mem) [36]. The samtools software (https://www.htslib.org/ 15 November 2025) was used to convert alignments from sam to bam formats, and Picard tools [37] were used to add read groups and sample IDs. Further processing was performed using the genome analysis tool kit (GATK v4.6) HaplotypeCaller to generate the genome variant call format (GVCF) files for each sample. The gvcf files were combined using GenomicsDBImport for variant calling using genotype GVCFs from GATK v4.6 [38]. The resulting variant call file was pre-filtered for minor allele frequency (>0.05), genotype missingness (<15%), and insertions and deletions (INDELs) using vcftools v0.1.13. Additional filtering was performed in Tassel (v5.2.52) at 0.05 minor allele frequency to retain SNPs that were used in the downstream analyses. Data was then imputed using K nearest neighbor genotype imputation method [39], and 61736 SNPs were used for downstream analysis. The SNP diversity was summarized by calculating the distribution of minor allele frequency (MAF) [39], polymorphic information content and heterozygosity (PIC) [40]. Sparse non-negative matrix factorization (SNMF) was used to create genetic clusters of the strains for different values of K ranging from K = 1 to K = 10 to find the optimum number of clusters that can explain the genetic variation [41]. A cross-entropy plot was generated to identify the minimum number of clusters that can explain the variation between clusters. Principal component analysis (PCA) and discriminant analysis of principal component (DAPC) were also used to determine the population structure of the strains. The optimum number of genetic clusters was further verified in Structure software v2.3.4, according to procedure by Evanno et al. [42].

Analysis of molecular variance (AMOVA) was conducted to assess the variation among the strains based on the genetic clusters using “poppr” and “adegenet” packages in R software v4.5.2 [43,44]. Genetic variation was portioned into variation within and between genetic clusters. A phylogenetic tree was generated using Euclidean distance matrix in PowerMarker v3.25 [40] and visualized using fig tree v1.4.4. Pairwise Wright’s fixation indices (Fst) was computed between the different genetic clusters to determine the genetic differentiation among the clusters. Association between the clusters, geographical origin, pathogenicity, growth rate, and number of sclerotia were tested using Chi square (X).

## 3. Results

### 3.1. Morphological Diversity of Sclerotium rolfsii Strains

Data were collected on the growth rate, colony texture, and number of sclerotia for the 84 strains obtained in 2020 and 2021. The growth rate of the strains ranged from 1.1 to 3.6 cm/day, while the number of sclerotia produced ranged from 0 to 543. The number of sclerotia produced was categorized as none, low, medium, and high as follows: 0 = none, 1 to 49 = low, 50 to 99 = medium, and >100 = high. A total of 12, 21, 20, and 31 strains were in the groups of high, medium, low, and none, respectively. The strains with cottony colony texture (Figure 2) were classified as fibrous (12 strains), compact (two strains), and fluffy (70 strains). The sclerotia were brown for 40 strains and dark brown for 13 strains, while 31 strains did not form sclerotia by day 28 post-inoculation (Appendix A). Significant differences were observed in the number of sclerotia produced by strains sampled from different agro-ecological zones (F = 5.6, *p* = 0.001). Following Tukey’s HSD test, the average number of sclerotia produced by strains from Lake Victoria Crescent and Mbale farmlands (LVC), Northern Mixed Farming System (NMFS), West Nile Mixed Farming System (WNFS), and Tanzania were similar, while that of the rest of the agro-ecological zones differed (Figure 3A). Meanwhile, strains from Eastern Highlands (EH) had the highest average sclerotia number of 156, and the ones from the Teso Farming System Zone (TFZ) had the lowest average number of sclerotia.

Additional data on the rest of the strains (104 collected in 2013) was obtained from Paparu et al. (2020) [9] and summarized in Appendix A. Significant differences were also observed in the average growth rate of the strains from distinct sampling locations (F = 12, *p* = 0.0001) (Figure 3B). The strains from WNFS and TFZ had similar average growth rates, while the strains from the rest of the agro-ecological zones differed in the growth rates. The strains from Bukoba in Tanzania had the highest average growth rate of 2.81 ± 0.054 cm/day, while those from EH had the lowest growth rate of 2.14 ± 0.058 cm/day.

**Figure 2 jof-12-00018-f002:**
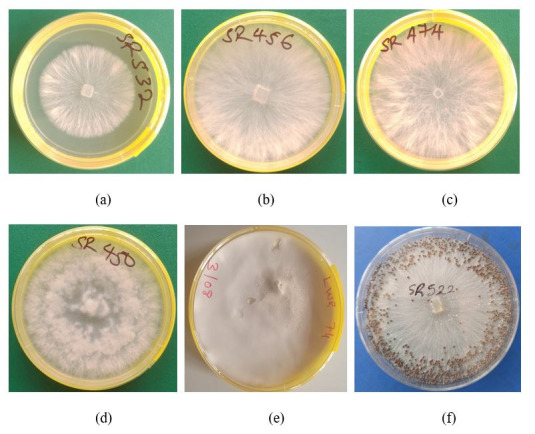
Colony characteristics of *Sclerotium rolfsii* strains following culture on potato dextrose agar (PDA). (**a**–**c**). Colonies produced by strains (SR532, SR456, and SR474) with the fibrous texture. (**d**) Colony produced by strain SR450 with the fluffy texture. (**e**) Colony produced by strain LWR74 with compact texture. (**f**) Numerous sclerotia produced by strain SR522 in culture.

**Figure 3 jof-12-00018-f003:**
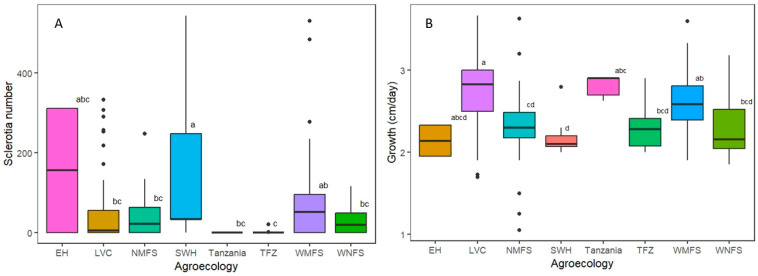
Comparison of the number of sclerotia and growth rates among the strains sampled from distinct geographic locations. (**A**) The average number of sclerotia. (**B**) The growth rate of different *S. rolfsii* strains and their agro-ecological zones of origin. Agro-ecological zones with different letters are significantly different EH = Eastern Highlands, LVC = Lake Victoria Crescent and Mbale farmlands, NMFS = Northern Mixed Farming System, SWH = South-Western Highlands, TFZ = Teso Farming Zone, WMFS = Western Mixed Farming System, and WNFS = West Nile Mixed Farming System.

### 3.2. Pathogenic Diversity of Sclerotium rolfsii

#### 3.2.1. Virulence of *S. rolfsii* Strains

Pathogenicity tests were conducted on 188 strains. Of these, 104, 12, and 72 were collected in 2013, 2020, and 2021, respectively (Appendix A). Disease Severity Index (DSI) for the strains was determined separately for each experiment. There were no significant differences in the DSI among *S. rolfsii* strains in experiments 1, 3, and 5 (F = 1.2532, *p* = 0.5422, F = 2.4081, *p* = 0.2561, F = 2.7442, and *p* = 0.3033 for experiments 1, 3, and 5, respectively), while there were significant differences in the DSI among strains in experiments 2 and 4 (F = 6.6546, *p* < 0.0001 and F = 8.4175, *p* < 0.0001 for experiments 2 and 4, respectively). The coefficient of determination (R^2^) was calculated for the strains and tray, which was used as a random factor. The R^2^ for the strain effects were 0.28, 0.42, 0.18, 0.44, and 0.38 for experiments 1 to 5, respectively. Meanwhile, the R^2^ for trays were 0.49, 0.42, 0.81, 0.48, and 0.48 for experiments 1 to 5, respectively. The average R^2^ for isolates and trays across all the experiments were 0.407 and 0.6731, respectively.

Overall, the most virulent strain was SR282 from the West Nile Mixed Farming System (WNFS), with an average DSI of 93.3%. The least virulent strains were SR475 and SR489, both from Lake Victoria Crescent and Mbale farmland (LVC), with an average DSI of 10.1%. Out of the 10 most pathogenic strains, 8 were from the 2013 collection, while only two were from the 2020 and 2021 collections (Table 1 and Appendix A). The average DSIs of 2013, 2020, and 2021 were significantly different from each other (F = 17.2, *p* = 0.001). The average DSI of 2013, 2020, and 2021 strains across all the experiments were 59.5%, 46.3%, and 48.8%, respectively. Out of the 10 least virulent strains, 7 were from the 2020 and 2021 collections while only three were from the 2013 collection (Table 2 and Appendix A). The average DSI of strains from the different agro-ecological zones in Uganda ranged between 30.6% for Eastern Highland (EH) to 63.9% for South-Western Highland (SWH) (Table 3). The average DSI of strains from Bukoba in Tanzania was 76.8%, which was the highest among all the geographical locations (Appendix A). Chi-square revealed no association between year of storage and virulence (X = 329, *p* = 0.351).

When the strains were grouped according to their origin, there were no significant differences in their virulence in experiment 1 (F = 1.5089, *p* = 0.2), experiment 4 (F = 1.1.3089, *p* = 0.2792), and experiment 5 (F = 0.4489, *p* = 0.3968). Strains evaluated in experiments 1, 4, and 5 originated from the Eastern Highland (EH), Lake Victoria Crescent and Mbale farmland (LVC), Northern Mixed Farming System (NMFS), South-Western Highlands (SWH), Western Mixed Farming System (WMFS), and West Nile Mixed Farming System (WNFS). However, significant differences were observed in the DSI for strains originating from different agro-ecological zones in experiment 2 (F = 3.4493, *p* < 0.0118) and experiment 3 (F = 6.1513, *p* < 0.0004). A pairwise comparison of DSI among the different agro-ecological zones was conducted. In experiment 2, there were significant differences in the DSI between strains from the Teso Farming Zone (TFZ) and those from LVC (*p* = 0.013), NMFS (*p* = 0.009), SWH (*p* = 0.014), and NMFS (*p* = 0.036). Meanwhile, in experiment 3, there were significant differences in the DSI of strains from LVC and strains from Tanzania. Overall, significant differences were observed between strains from EH, NMFS, and Tanzania (Appendix A). The R^2^ values for agro-ecological zones were 0.03, 0.05, 0.19, 0.04, and 0.01 in experiments 1 to 5, respectively, while the R^2^ for the trays were 0.34, 0.4, 0.52, 0.45, and 0.35 in experiments 1 to 5, respectively. For all strains obtained from Uganda, the strains from SWH had the highest average DSI, while those from TFZ had the lowest average DSI. However, the highest DSI for all *S. rolfsii* strains was observed for strains from Tanzania.

**Table 3 jof-12-00018-t003:** Average Disease Severity Index (DSI), growth rate, and the number of sclerotia produced by *Sclerotium rolfsii* strains from distinct agro-ecological zones.

S/No	Agro-Ecological Zones	Number of Strains	DSI (%)	Growth Rate (cm/Day)	Sclerotia Number
1	EH	2	30.3 ± 2.2	2.14 ± 0.058	66.3
2	LVC	79	51.9 ± 2.1	2.75 ± 0.052	59.7
3	NMFS	24	46.7 ± 2.2	2.31 ± 0.053	59.4
4	SWH	10	63.9 ± 1.9	2.18 ± 0.053	59.9
5	Tanzania	2	76.8 ± 1.9	2.81 ± 0.054	59.6
6	TFZ	10	36.7 ± 2.0	2.18 + 0.053	57.2
7	WMFS	46	64.4 ± 1.2	2.31 ± 0.053	59.6
8	WNFS	15	49.1 ± 2.4	2.34 ± 0.053	59.1

EH—Eastern Highlands, LVC—Lake Victoria Crescent and Mbale farmland, NMFS—Northern Mixed Farming System, SWH—South-Western Highlands, TFZ—Teso Farming Zone, WMFS—Western Mixed Farming System, and WNFS—West Nile Mixed Farming System.

There was a weak negative correlation between the growth rate and pathogenicity (r = −0.06) and the number of sclerotia produced (r = −0.02). A weak positive correlation was observed between pathogenicity and the number of sclerotia produced (r = 0.06)

#### 3.2.2. Susceptibility of Common Bean Varieties to Southern Blight

In experiment 3, there were no significant differences in DSI for the five common bean varieties screened (F = 2.4081, *p* = 0.2561). However, DSI was significantly different among common bean varieties in experiments 1 (F = 3.555, *p* = 0.007) 2 (F = 8.242, *p* = 0.02), 4 (F= 15.677, *p* < 0.0001), and 5 (F= 6.2736, *p* < 0.0001). Pairwise comparison of *p* values among the varieties within each experiment was conducted. In experiment 1, the DSI of CAL96 varied with that of KWP9, while in experiment 2, that of CAL96 and WKP9 and MLB49-89A and RWR719 differed. In experiment 4, the DSI of NABE14 and RWR719, NABE14 and KWP9, MLB49-89A and RWR719, MLB49-89A and KWP9, and CAL96 and RWR719 differed. In experiment 5, the DSI of NABE14 and RWR719, NABE14 and KWP9, and CAL96 and KWP9 differed (Table 4). A heat map was then generated for DSI caused by the different strains (Figure 4a). Overall, CAL96 was the most susceptible variety, with an average DSI of 54.2%, and RWR719, the most resistant, with an average DSI of 42.4%. The R^2^ values for variety were 0.02, 0.02, 0.03, 0.12, and 0.05 for experiments 1 to 5, respectively, while the R^2^ values for the tray effect were 0.32, 0.43, 0.36, 0.56, and 0.39 for experiments 1 to 5, respectively. When the data for all the experiments was pooled, there were significant differences in DSI on the different common bean varieties (F = 9.205, *p* < 0.0001). A pairwise comparison of *p* values revealed that the DSI of CAL96 and RWR719 differed.

We also assessed seed germination to determine the extent of pre-emergence damping-off caused by the different *S. rolfsii* strains. In experiments 2, 4, and 5, significant differences were observed in the germination of the five common bean varieties following inoculation with *S*. *rolfsii* strains (F = 4.522, *p* <0.00001; F = 6.873, *p* < 0.0001; and F = 6.2736, *p* < 0.0001 for experiments 2, 4 and 5, respectively). Meanwhile, in experiments 1 and 3, no significant differences were observed in percentage germination following inoculation with *S*. *rolfsii* strains (F = 2.378, *p* = 0.05175 and F = 1.4193, *p* = 0.2293 for experiments 1 and 3). It is important to note here that each experiment had a different set of *S. rolfsii* strains. Overall, the lowest percentage germination of 22% was observed in experiment 2 following inoculation with strain SR31 from Kisoro district in SWH agro-ecology. The highest overall germination percentage of 100% was observed after inoculation with strain SR16 from Gulu district in the NMFS, still in experiment 2. A heat map of percentage of germination of common bean varieties against the strains is shown in Figure 4b. We observed a negative correlation between DSI and germination percentage across strains (r = −0.58). Inoculation with strains with higher DSI values resulted in lower germination percentages in the common bean varieties screened.

Average percentage germination was determined for the five varieties used in this study to assess the effect of varietal differences on pre-emergency damping-off. Across all the experiments, the average germination percentage of the varieties were 57.7, 58.9, 63.3, 65.2, and 69.46% for MLB48-89A, CAL96, NABE14, KWP9, and RWR719, respectively. Average germination percentages in the non-inoculated controls were 89.8, 92.28, 94.5, 94.5, and 95.3 for MLB48-89A, CAL96, NABE14, KWP9, and RWR719, respectively. A pairwise comparison of *p* values for germination was conducted among the different common bean varieties within each experiment. Significant differences were observed between varieties in experiments 2, 4, and 5. Overall, significant differences were observed in germination percentages of KWP9, MLB49-89A, and RWR719 (Table 5). The R^2^ values for varieties were 0.01, 0.02, 0.02, 0.12, and 0.05, respectively, for experiments 1 to 5, while those for the tray effects were 0.35, 0.43, 0.25, 0.51 and 0.39, respectively, for experiments 1 to 5. When data was pooled for all the experiments, significant differences were observed in percentage germination among the five common bean varieties (F = 5.9, *p* < 0.0001). The percentage of germination of MLB49-89A (57.5%) was significantly different from that of RWR719 (69.4%) and KWP9 (65.1%).

### 3.3. Genetic Diversity of Sclerotium rolfsii

One hundred sixty-one strains were used for genetic diversity studies. Following sequencing, the libraries generated a total of 2,921,752 reads. The sequences were deposited at the sequence read archive under accession SRX31254156 to 31,254,315 (Appendix A). The generated reads were mapped to the haploid reference genome assembly for isolate GP3 (Yan et al., 2021 [22]). After the removal of insertions and deletions and filtering at 0.95 missingness and 0.05% minor allele frequency, 61736 SNPs were retained for downstream analysis (Table 6). The polymorphic information content (PIC) ranged from 0.01227 to 0.37499, with an average of 0.25873.

#### Population Structure and Variation in Morphological Traits Between Groups

The population structure of the *S. rolfsii* strains was analyzed using principal component analysis and discriminant analysis of principal components. The optimum number of clusters for grouping *S. rolfsii* strains was determined to be 5 based on a plot of Bayesian information content (BIC) and discriminant analysis of principal components (Figure 5a,b and Appendix A). There were 12, 22, 31, 38, and 58 strains in clusters 1 to 5, respectively. The clusters to which the strains belong are presented in Appendix A. The strains did not cluster according to their agro-ecology of origin, with all clusters having strains from different agro-ecological zones.

We assessed the proportions of ancestry shared among the *S. rolfsii* strains. The number of sub-population (K) varied from 2 to 5, as per the procedure by Kopelman et al. [45]. All the strains had a shared ancestry with a phi of less than 1 (Appendix A). Pairwise Wright’s fixation indices (Fst) were used to assess the genetic differentiation among the strains from the different genetic clusters. The Fst ranged from 0.312 between genetic clusters 3 and 4 to 0.778 between genetic clusters 2 and 4, with an average of 0.5143 (Appendix A).

Following discriminant analysis of principal components, the different genetic clusters were separated by different axes, implying that they are genetically distinct (Figure 6a). A bar plot was generated from the Q-matrix to visualize the admixture among the *S. rolfsii* strains (Figure 6b). An analysis was conducted using the Adegenet package in R studio to identify significant SNPs associated with the genetic clusters. A loading plot was generated, and two significant SNPs on chromosome 3 (**Chr3_3160729**) and (**Chr3_2484656**) were identified (Figure 6c). The SNP allele frequencies were computed to determine whether there were changes in allele frequencies among the different genetic clusters. In the genetic clusters 2, 4 and 5, the frequency of adenine was 0 while that of thymine was 1 for both loci. In genetic clusters 1 and 3 there were changes in the allele frequency of adenine and thymine ranging from 0.06000 to 0.94000 for both loci (Appendix A).

Analysis of variance was conducted to compare variation in the morphological and virulence traits among these five genetic clusters. There were significant differences in the DSI caused by strains from different genetic clusters (F = 2.835, *p* = 0.0248) (Appendix A). The DSI of strains from clusters 2, 3, and 5 were significantly different, while that of cluster 1 and 4 were similar. The growth rate of strains in different clusters also varied significantly (F = 4.152, *p* = 0.0026). The strains from clusters 2, 3, and 4 had similar growth rates, while the growth rate of strains from clusters 1 and 5 differed significantly. The number of sclerotia produced by the strains from different clusters was significantly different, too (F = 14.6, *p* < 0.0001). The number of sclerotia produced by strains from cluster 5 was significantly different from the rest of the clusters (Appendix A). There was a significant association between the genetic clusters and number of sclerotia (X^2^ = 834.3, *p* = 0.00001), growth rate (X^2^ = 639, *p* = 0.000013), and DSI (X^2^= 1346, *p* = 0.00001).

Analysis of molecular variance (AMOVA) was conducted to assess proportions of genetic variance within and among groups of strains defined by their genetic relatedness (Table 7). Results revealed that 53.06999% of the variation was between populations, while 46.93001% of the variation was between strains within the population.

## 4. Discussion

### 4.1. Morphological Characterization of Sclerotium rolfsii Strains

We characterized 84 newly collected strains of *S. rolfsii* not previously characterized by Paparu et al. [10] for growth rate and number of sclerotia produced when grown on potato dextrose agar at 25 °C. *Sclerotium rolfsii* is known for its prolific growth rate in vitro. In the current study, the growth rate ranged from 1.1 to 3.6 cm per day. These growth rates are within the ranges reported in previous studies. For example, Paparu et al. [10] obtained growth rates of 1.04 cm to 2.92 cm per day on PDA, while growth rates reported by Le et al. [16] ranged from 0.67 to 1.8 cm per day. Paul et al. [46] studied the growth rate of *Athelia rolfsii* causing stem rot in peanuts and reported a high growth rate of 3.6 cm per day on PDA and lower growth rates on oatmeal agar (3.41 cm per day) and corn meal agar (2.61 cm per day). The growth rate of 66 *Athelia rolfsii* isolates in China ranged from 1.06 to 2.06 cm per day [47]. Paul et al. [46] showed that growth rates depend on temperature, with the highest and lowest growth rates observed at 30 °C and 15 °C, respectively.

In the current study, there was a weak negative correlation between growth rate and pathogenicity and a weak positive correlation between pathogenicity and the number of sclerotia produced. Similarly, Paparu et al. [10] reported moderate positive correlations between pathogenicity and the number of sclerotia produced, while Yan et al. [47] reported no significant differences in the growth rates of highly and weakly aggressive strains of *S. rolfsii* causing peanut stem rot. Other studies showed a positive correlation between pathogenicity and the number of sclerotia produced [47,48]. Comparison of the number of sclerotia produced in different studies shows that *S. rolfsii* strains are capable of producing a high number of sclerotia. In peanuts, 35 strains produced less than 100 sclerotia per Petri dish, while 25 strains produced more than 300 [47]. In our study, 75 strains did not produce sclerotia by day 28. In a related study by Xie et al. [49], only 1 strain out of 19 failed to produce sclerotia. However, we observed that, when the cultures were kept for more than two months, all the strains eventually produced sclerotia, suggesting that all strains of *S. rolfsii* are likely able to produce sclerotia under conditions of nutrient stress. This is compatible with the fact that the sclerotia is known to be a fruiting body that is used by the pathogen for transition from one season to another, even in the absence of the host plant [8].

### 4.2. Pathogenicity of Sclerotium rolfsii Strains on Common Bean

We carried out pathogenicity tests on 188 *S. rolfsii* strains using five common bean varieties. Importantly, we found that strains stored for over 10 years were still virulent on common bean and there was no association between virulence and years of storage. The ability to store strains for long periods of time opens opportunities for creating collections of strains, a resource that is critical for (1) improving the pathogen surveillance system, (2) studying pathogen population dynamics over time, (3) studying pathogen adaptation to environmental changes, and (4) conducting virulence studies to identify resistant crop varieties. All analyzed *S. rolfsii* strains were virulent on common bean and caused all the symptoms associated with southern blight such as brown water-soaked lesions at the collar of seedling, presence of white cottony mycelia on lesions, wilting of plants, presence of sclerotia at the collar of plants, and pre-emergence and post-emergence dumping-off. Some strains caused severe disease while others caused mild disease. Similarly, the strains also differed in their effect on the germination of common bean varieties, with some strains causing more pre-emergence damping-off than others. The high Disease Severity Index (DSI) was associated with the high frequency of pre-emergence damping-off.

In the current study, the strains from different agro-ecological zones differed in virulence. The most pathogenic strains were from Northern Mixed Farming System (NMFS), followed by South-Western Highlands (SWH), the West Nile Mixed Farming System (WNFS), Lake Victoria Crescent and Mbale farmlands (LVC), the Western Mixed Farming System (WMFS), Eastern Highlands (EH), and Teso Farming Zone (TFZ), respectively. *Sclerotium rolfsii* strains from Bukoba in Tanzania had the highest average DSI compared to all Ugandan strains. This observation may have serious implications for the spread of these highly virulent strains to other regions of East Africa, because traders often transport grains across locations, and farmers use these grains as seed sources. According to WITS [50], Uganda imported 25,569.1 tons of common bean from Tanzania in 2023. Since the common bean seed system in Uganda is not very well developed, farmers end up using these grains and those from the previous season as seeds (MAAIF) [51]. Coyne et al. [52] reported infected seeds as a major factor contributing to the spread of common bean diseases. Since *S. rolfsii* mostly reproduces asexually, these strains can rapidly multiply in their new locations. Similar to our findings, in lentil, *S. rolfsii* isolates from different agro-ecological zones also showed differences in pathogenicity [53].

DSI of the analyzed strains ranged from 10.1 to 93.3%, which is comparable to the previous report by Paparu et al. [10], where the DSI ranged from 4.4 to 100%. The DSI caused by different strains on common bean also varied across varieties, with RWR719 being the most resistant and CAL96 the most susceptible variety. The above findings are similar to those reported by Paparu et al. [10]. Other studies have reported variation in the pathogenicity of *S. rolfsii* strains in various crop varieties [54,55,56]. Paul et al. [54] reported variations in the virulence of *S. rolfsii* on different sweet potato varieties, concluding that different races of *S. rolfsii* might exist in South Korea. In a related study, eight *S. rolfsii* isolates were used to screen five varieties of eggplants and tomatoes; all the varieties were graded as susceptible with mortality of up 93% for some isolates [56].

*Sclerotium rolfsii* has a broad host range and causes disease in various crops. Paparu et al. [10] obtained a strain HOI356-Peanut from a volunteer groundnut plant in the bean field that was also pathogenic on the common bean. In the current study, the strain was renamed SR411 and caused a DSI of 34.5%. In the pathogenicity studies of *S. rolfsii* obtained from groundnuts, tomatoes, and taro, strains from tomatoes and taro were found to be pathogenic on groundnuts [16]. Nineteen *S. rolfsii* strains collected from various crops in the Southern United States (Florida, Georgia, Louisiana, South Carolina, Texas, and Virginia) were also pathogenic on tomatoes, peppers, and peanuts [49]. Paul et al. [46] used two *S. rolfsii* strains from common bean to conduct pathogenicity on other crops. They observed that both strains were pathogenic on tomatoes, eggplants, and chickpeas but not on chilis, soybeans, and cowpeas. In spite of a wide host range, not all strains are pathogenic across all *S. rolfsii* hosts. This means that resource-constrained farmers need to implement appropriate crop rotation strategies by including non-host crops to reduce inoculum levels in the soil if they are to manage southern blight through crop rotation [14].

### 4.3. Molecular Diversity of S. rolfsii Strains

Various molecular diversity parameters were computed to determine the diversity among the strains. The average polymorphic information content (PIC) of *S. rolfsii* strains was moderate. This is consistent with a study involving *S. rolfsii* causing collar rot of eggplant and tomato, where the PIC per chromosome ranged from 0.145 to 0.525 with an average of 0.322 [52]. While studying the genetic diversity of *S. rolfsii* infecting various crops in China using RAPD, Wang et al. [56] observed a polymorphic information content greater than 0.5. Meena et al. [57] observed a polymorphic information content of 0.242 to 0.536 among *S. rolfsii* infecting groundnuts while using RAPD. These findings show that the genetic diversity among *S. rolfsii* is moderate, which can reduce the cost of breeding since not so many strains have to be used for the screening process.

Analysis of population structure in the current study indicates that strains could be optimally clustered into five genetically distinct groups. The population of *S. rolfsii* from the Ugandan agro-ecological zone showed evidence of admixture with the strains from other agro-ecological zones. The strains from these five genetic groups showed significant differences in the DSI, number of sclerotia produced, and the rate of growth. It is not immediately clear whether these differences have adaptive value or not, because no strong correlation between virulence of the common bean and these morphological and pathogenicity characteristics of the strains was detected. Meanwhile, in the study by Yan et al. [47], while clustering isolates of *Athelia rolfsii* causing stem rot in peanuts in China using ISSR markers, the isolates clustered based on geographical origin and sclerotia traits but not virulence. In the current study, clustering was not by agro-ecological zone of origin. Contrary to these results, the clustering of *S. rolfsii* strains pathogenic to groundnuts using sequence data from the ITS of ribosomal DNA was consistent with the strains’ geographic origins [16].

The pairwise genetic distance between genetic clusters based on Wright’s fixation index ranged from 0.312 to 0.770 between clusters with an average of 0.5143, which is a moderate value. Meena et al. [57] used ITS and RAPD primers to study genetic diversity among *S. rolfsii* infecting groundnuts. In the study, the similarity coefficient ranged from 0.33 to 0.57 between different isolates. While using RAPD markers to study genetic diversity among *S. rolfsii* causing collar rot in chickpeas, Srividya et al. [58] observed a similarity index of 0.15 to 0.72 and dissimilarity index of 0.75 to 0.83. They concluded that the populations of this fungus are made up of more than one race and that few are derived clonally.

Following AMOVA, the 53% proportion of variance explained by groups indicates a moderate population structure in the sampled collection of *S. rolfsii* strains. The 47% proportion of genetic variance attribute to within population suggests a moderate level of inter-mating among *S. rolfsii* strains in the collection. These results indicate very moderate levels of genetic differentiation among populations, which could be associated with substantial inter-population gene flow or shared ancestry. The moderate values of genetic variance between populations (φ = 0.5399531) suggest that about 50% genetic variance is between populations rather than within populations. This result could indicate that there are moderate levels of inter-mating among strains within the sampled geographic range or there is long-distance movement of strains. The latter could be facilitated by the widespread pulse movement across the country, as farmers often purchase them and use them as seeds for the next planting season. While analyzing population structure of *S. rolfsii* strains infecting Chinese herbal crops, Wang et al. [56] identified three groups, and the differences between the groups were mostly related to geographical location. The analysis of molecular variance showed that 28% of the variation was among the groups, while 72% was within the groups. In their study, cluster analysis showed that the closely spaced populations grouped in one cluster, which implies that the isolates in their study may not have been highly mobile. In the study by Mehri et al. [59], 70% and 30% of the observed variance corresponded to the difference between and within the populations, respectively. They suggested that mating between populations would be less likely, and thus, gene flow is restricted.

An analysis was conducted to identify significant SNPs associated with the observed genetic clusters/populations. Two significant SNPs were identified on chromosomes 3. Further analysis was conducted to determine the variation of allele frequency at the responsible loci. The allele frequencies varied among the genetic clusters. Grünwald et al. [60] attributed changes in allele frequency among pathogen populations to several factors such as mutation, gene flow, genetic drift, and sexual recombination. They suggested that such changes may lead to the production of a more virulent population. The virulent pathogens increase in frequency through directional selection and may spread to neighboring areas or continents through natural or human-mediated gene flow [61].

*Sclerotium rolfsii* causing southern blight of the common bean is a morphologically and genetically diverse pathogen. Significant differences in morphological and virulence traits have been detected among genetically differentiated groups of strains. The collected *S. rolfsii* strains showed moderate levels of genetic diversity. Quantifying genetic variance at different levels of population hierarchy showed that the proportion of genetic variances explained by variation within the population and between populations are almost equal. Since there are no regionally specific virulent strains of the *S. rolfii*, new resistant varieties developed and evaluated in one region could easily adapt to other regions within the country, reducing the burden to breed region-specific varieties.

## 5. Conclusions

*Sclerotium rolfsii* causing southern blight of the common bean is a morphologically and moderately genetically diverse pathogen. Significant differences in morphological and virulence traits have been detected among genetically differentiated groups of strains. The collected *S. rolfsii* strains showed moderate levels of genetic diversity. Our study shows a moderate level of population stratification in *S. rolfsii* and moderate genetic differentiation among strains from distinct geographic regions. The moderate levels of genetic diversity in *S. rolfsii* suggest that multiple strains are required for adequate screening of germplasm to identify novel sources of resistance. Since there are no regionally specific virulent strains of *S. rolfsii,* new resistant varieties of common bean developed and evaluated in one region could easily adapt to other regions within the country, reducing the burden to breed region-specific varieties. The movement of common bean seeds across Uganda and the re-use of these seeds in subsequent production seasons by farmers likely had a significant impact on the genetic diversity of the *S. rolfsii* population. To prevent the introduction of virulent strains from Tanzania into Uganda, better control of the movement of bean grain and seed need to be implemented.

## Figures and Tables

**Figure 1 jof-12-00018-f001:**
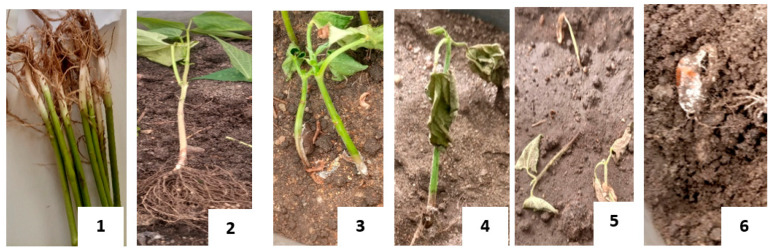
Southern blight Disease Severity Index scale (1–6): 1 = no symptoms, 2 = gray water-soaked lesions on stem above soil but no visible fungal growth, 3 = visible fungal growth at the base of the stem near the soil line, characterized by silky white mycelia or sclerotia that gradually darken, 4 = partial wilting, where young leaves begin to wilt and stems begin to shrivel, 5 = complete wilting, desiccation, and browning of leaves and stem collapse and death of plant, and 6 = pre-emergence damping-off, complete seed death characterized by rotting and fungal growth.

**Figure 4 jof-12-00018-f004:**
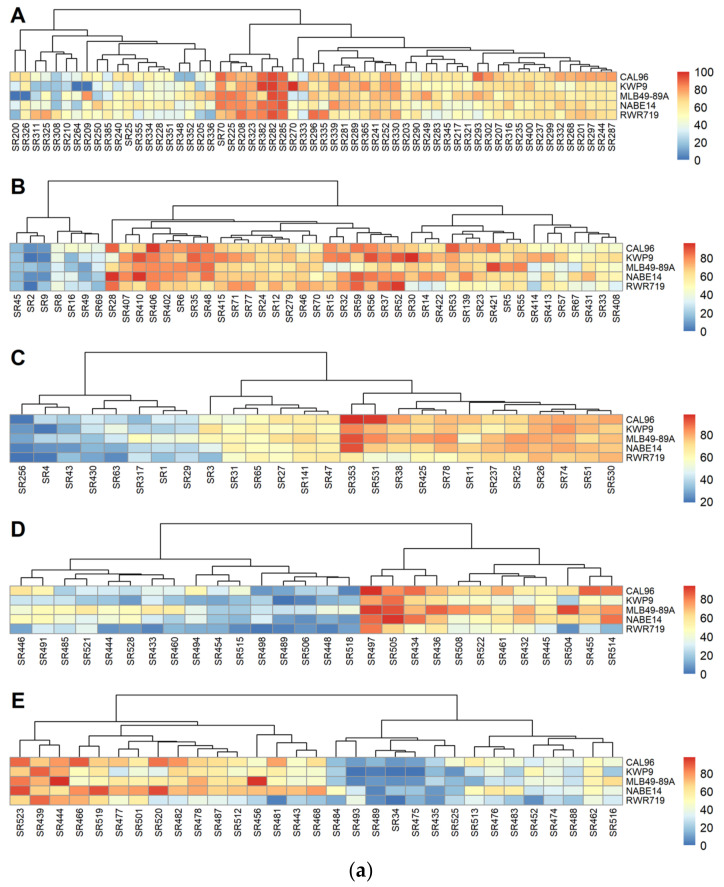
(**a**) The heat map of DSI of different *S. rolfsii* strains. A to E are for experiments 1, 2, 3, 4, and 5, respectively. (**b**) The pre-emergency damping-off heat map of different *S. rolfsii* strains against the five common bean varieties. A to E are for experiments 1, 2, 3, 4, and 5, respectively.

**Figure 5 jof-12-00018-f005:**
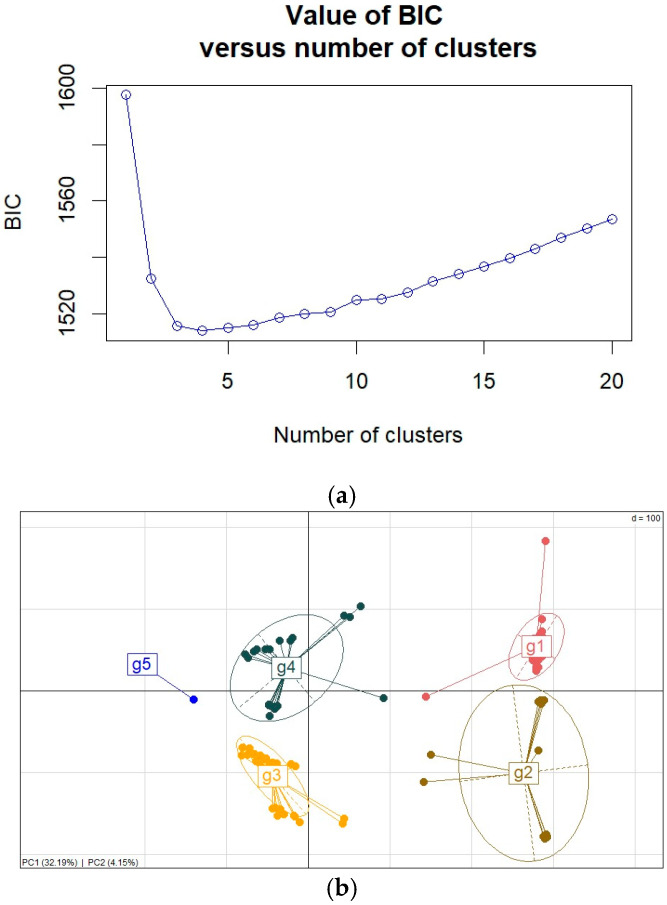
(**a**) Bayesian information content plot, (**b**) clustering of *S. rolfsii* strains using principal component analysis based on two principal components.

**Figure 6 jof-12-00018-f006:**
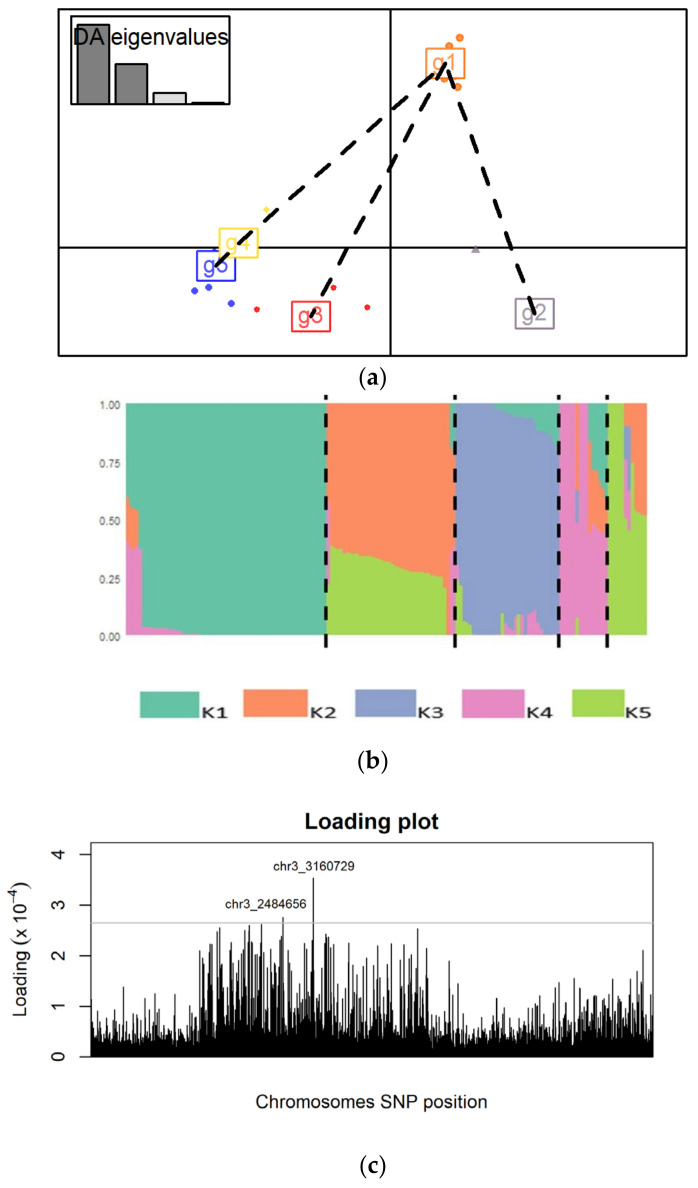
(**a**) A scatter plot of the different genetic clusters showing separation of the clusters by the different axes, (**b**) proportions of *S. rolfsii* strains ancestry assigned to K populations, and (**c**) the loading plot showing SNPs associated with the genetic clusters. The loading threshold was set at 0.00027.

**Table 1 jof-12-00018-t001:** Average DSI of 10 most-virulent *Sclerotium rolfsii* strains screened against five common bean varieties in the screen house.

S/No	Strain	Agro-Ecology ^1^	District	Year	DSI	SE
1	SR282	WNFS	Arua	2013	93.4	8.6
2	SR417	TFZ	Bukedea	2013	91.7	4.2
3	SR406	LVC	Luweero	2013	84.5	2.2
4	SR70	LVC	Mubende	2013	84.3	7.8
5	SR497	NMFS	Oyam	2020	82.2	1.5
6	SR52	LVC	Luweero	2013	81.8	1.5
7	SR505	LVC	Kamuli	2020	80.9	0.7
8	SR59	LVC	Mbale	2013	80.6	2.9
9	SR56	LVC	Mbale	2013	80.3	1.5
10	SR410	LVC	Mbale	2013	80.1	1.9

^1^ WNFS = West Nile Mixed Farming System, TFZ = Teso Farming Zone, LVC = Lake Victoria Crescent and Mbale farmland, and NMFS = Northern Mixed Farming System.

**Table 2 jof-12-00018-t002:** Average DSI of 10 least-virulent strains when screened against five common bean varieties in the screen house.

S/No	Strain	Agro-Ecology ^1^	District	Year	DSI	SE
1	SR45	LVC	Luweero	2013	18	2.9
2	SR493	LVC	Sironko	2021	16.8	2.1
3	SR435	WMFS	Hoima	2021	16.1	0
4	SR454	WMFS	Hoima	2021	16	1.9
5	SR264	LVC	Bugiri	2013	15.9	1.5
6	SR518	LVC	Sironko	2021	15.5	1.9
7	SR506	LVC	Lwengo	2020	13.2	1.5
8	SR9	NMFS	Apac	2013	10.2	0.1
9	SR489	LVC	Sironko	2021	10.1	1.4
10	SR475	LVC	Sironko	2021	10.1	1.1

^1^ LVC = Lake Victoria Crescent and Mbale farmland, WMFS = Western Mixed Farming System, and NMFS = Northern Mixed Farming System.

**Table 4 jof-12-00018-t004:** The DSI on different common bean varieties across the different experiments.

S/No	Variety	Expt 1	Expt 2	Expt 3	Expt 4	Expt 5	Average
1	CAL96	45.5 ± 1.8 **	60.8 ± 3.6	65.9 ± 4.7 **	49.2 ± 5.4	51.7 ± 5.3 **	54.2 ± 3.9 **
2	KWP9	33.2 ± 2.5 **	65.7 ± 3.9	62.4 ± 4.7 **	33.4 ± 5.1 **	40.8 ± 5.1 **	46.9 ± 3.4
3	MLB49-89A	40.5 ± 2.4	65.3 ± 3.2	68.3 ± 4.4 **	55.6 ± 5.5 **	45.1 ± 5.4	54.3 ± 3.9
4	NABE14	42.1 ± 2.4	61.9 ± 3.8	62.9 ± 4.9	46.2 ± 5.2 **	54.3 ± 5.6 **	53.1 ± 3.8
5	RWR719	35.5 ± 3.0	56.2 ± 4.0	56.3 ± 5.3 **	28.1 ± 5.1 **	35.7 ± 5.0 **	42.4 ± 3.1 **

** Significant at 95% confidence interval.

**Table 5 jof-12-00018-t005:** Germination of common bean varieties across different experiments.

S/No	Variety	Expt 1	Expt 2	Expt 3	Expt 4	Expt 5	Average
1	CAL96	77.5 ± 3.9	50.5 ± 7.6	63.3 ± 6.6	64.6 ± 12.2 **	60.6 ± 5.6 **	63.2 ± 2.1
2	KWP9	80.3 ± 3.5	44.0 ± 6.5	64.9 ± 6.7	75.7 ± 14.3 **	65.1 ± 5.1 **	65.1 ± 2.1 **
3	MLB49-89A	73.0 ± 3.4	43.9 ± 6.5 **	55.8 ± 7.6	51.7 ± 9.6 **	64.7 ± 5.4 **	57.5 ± 2.0 **
4	NABE14	80.1 ± 3.5	50.0 ± 7.4	67.0 ± 6.7	64.8 ± 12.2 **	56.3 ± 5.5 **	63.2 ± 2.1
5	RWR719	78.7 ± 3.7	54.7 ± 8.0 **	71.1 ± 6.2	78.8 ± 14.9 **	69.3 ± 5.4 **	69.4 ± 2.1 **

** Significant at 95% confidence interval.

**Table 6 jof-12-00018-t006:** Total number of SNPs and SNPs retained after removing insertions and deletions and filtering at 0.95 missingness and 0.05 minor allele frequency and the polymorphic information content (PIC) per chromosome.

Chromosome	Total SNPs	Number of SNP’s	Average PIC
1	388,990	11,961	0.2540
2	399,572	225	0.2358
3	751,857	24,179	0.2691
4	185,670	455	0.2615
5	277,593	66	0.2328
6	287,247	128	0.3037
7	221,193	697	0.2611
8	409,630	24,025	0.2528
Overall	2,921,752	61,736	0.2587

**Table 7 jof-12-00018-t007:** Analysis of molecular variance outputs based on five optimal genetic clusters.

AMOVA Based on the Five Optimal Genetic Clusters					
Sources of Variation	Df	SS	MSS	%	Phi
Between populations	4	3,025,706	756,426.48	53.06999	1
Between samples within population	156	3,347,772	21,460.08	0	0
Between samples across population	156	3,347,772	21,460.08	46.93001	1
Total	321	6,373,478	1955.07	100	

## Data Availability

The sequences were deposited at the NCBI under BioProject accession PRJNA1370308 (Appendix A). The original contributions presented in this study are included in the article and Appendix A. Further inquiries can be directed to the corresponding author.

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
