# Peer review of "Pathogenic and Genetic Diversity of Sclerotium rolfsii, the Causal Agent of Southern Blight of Common Bean in Uganda"

_jof, 2025, doi:10.3390/jof12010018_

Round 1
Reviewer 1 Report
Please refer to the detailed comments.
Comments to the author (JOF-3865852)
The manuscript title, “Pathogenic and Genetic Diversity of Sclerotium rolfsii, the Causal Agent of Southern Blight of Common Bean in Uganda” by Erima et al is focusing on the interesting aspects of genetic and pathogenic diversity of S. rolfsii, fungal pathogen of common beans. I feel that this is an ambitious and valuable study and the authors combine morphological and pathogenic characterization of a large set of 188 Sclerotium/Athelia isolates with whole-genome re-sequencing (retaining ~51k SNPs after pruning), population structure analyses and link genetic clusters to morphological and virulence traits. The dataset and the attempt to connect genomics to phenotype are strengths and would be of interest to the community and plant breeders. Overall, the manuscript is well-structured and covers a comprehensive information on this pathogen. However, some concerns and suggestions from my side can be seen below for the consideration of authors and I am sure by addressing these concerns, the clarity and quality of the manuscript will be improved.
Section 2.2.3: In this section, each strain was inoculated in a single tray, replicated once, with 16 plants per line per tray. This design raises pseudoreplication concerns, because plants within a tray are not independent experimental units. The statistical model must account for the tray/replicate structure. Currently, the analysis treats plant-level DSI as independent, which is not valid. I would suggest re-analysing the data using a mixed model (tray as random effect) or aggregate at tray level.
Section 2.4.1: The DSI is derived from ordinal severity scores (1-6). The manuscript uses ANOVA and Tukey HSD, which assume continuous, normally distributed residuals. This may not be appropriate. I would recommend to justify assumptions with residual diagnostics or use an ordinal regression / GLMM approach.
Results: in the results section (420-425), read mapping ranged only 50-70% with some isolates <1× depth excluded (12 samples). This is unusually low for mapping to a reference genome and raises questions of contamination or high divergence. Please provide a per-sample QC table (% mapped, depth, missing data) and explain why mapping was low. Demonstrate that contamination was not an issue.
Further, in lines 430-432 and Table 4, the reported heterozygosity is 0.782, which is very high for a fungal pathogen. The pipeline used GATK HaplotypeCaller (default diploid assumptions), yet S. rolfsii is generally haploid/heterokaryotic. Treating isolates as diploid may inflate heterozygosity estimates. Please state expected ploidy of isolates, re-run or re-interpret SNP calling under correct assumptions, and provide allele frequency/allele balance plots to exclude artifacts or mixed cultures.
AMOVA shows 87.6% of variance “within samples” and only 0.39% between populations, yet the manuscript describes 5 genetic clusters as distinct. The high within-sample variance likely reflects heterokaryosis or analysis artifacts. Please clarify biological meaning of “within sample” variance. Reconcile AMOVA with clustering results. Avoid over-interpretation of weak structure.
Other than these concerns, the authors have used too many abbreviations, I would strongly recommended to use the complete name at first mention and later on, use only abbreviation. Importantly, standardize agro-ecological zone abbreviations and ensure they match in all tables/figures (EH, LVC, NMFS, SWH, TFZ, WMFS, WNMFS).
There are too many figures and tables, I would suggest shifting some figures and tables to supplementary data, figures up to 6 and tables up to 3-4/5 would be sufficient.
In the discussion section, only few references were cited, a strong discussion provides the support so, I would suggest expanding discussion based on the results of the present study.
Author Response
The manuscript has been attached.
Reviewer 1 comments to the author (JOF-3865852)
The manuscript title, “Pathogenic and Genetic Diversity of Sclerotium rolfsii, the Causal Agent of Southern Blight of Common Bean in Uganda” by Erima et al is focusing on the interesting aspects of genetic and pathogenic diversity of S. rolfsii, fungal pathogen of common beans. I feel that this is an ambitious and valuable study and the authors combine morphological and pathogenic characterization of a large set of 188 Sclerotium/Athelia isolates with whole-genome re-sequencing (retaining ~51k SNPs after pruning), population structure analyses and link genetic clusters to morphological and virulence traits. The dataset and the attempt to connect genomics to phenotype are strengths and would be of interest to the community and plant breeders. Overall, the manuscript is well-structured and covers a comprehensive information on this pathogen. However, some concerns and suggestions from my side can be seen below for the consideration of authors and I am sure by addressing these concerns, the clarity and quality of the manuscript will be improved.
Reply
Dear reviewer, thanks for your comments, they have been very valuable to us and have made us to realize some oversights which we have now addressed
Comment
Section 2.2.3: In this section, each strain was inoculated in a single tray, replicated once, with 16 plants per line per tray. This design raises pseudoreplication concerns, because plants within a tray are not independent experimental units. The statistical model must account for the tray/replicate structure. Currently, the analysis treats plant-level DSI as independent, which is not valid. I would suggest re-analysing the data using a mixed model (tray as random effect) or aggregate at tray level.
Reply
We used this design because the sample size was quite large, and the split-plot design can address this problem of less space. We decided to use a generalized linear mixed model and fitted tray number as a random factor
Comment
Section 2.4.1: The DSI is derived from ordinal severity scores (1-6). The manuscript uses ANOVA and Tukey HSD, which assume continuous, normally distributed residuals. This may not be appropriate. I would recommend to justify assumptions with residual diagnostics or use an ordinal regression / GLMM approach.
Reply
We have now used a generalized linear mixed model where we fitted tray numbers as a random factor. We then conducted ANOVA after running the GLMM and used pairwise comparison of P values to identify variables that were significantly different from each other. We also generated R² for the dependent variable and the random factor to check the effectiveness of the model
Comment
Results: in the results section (420-425), read mapping ranged only 50-70% with some isolates <1× depth excluded (12 samples). This is unusually low for mapping to a reference genome and raises questions of contamination or high divergence. Please provide a per-sample QC table (% mapped, depth, missing data) and explain why mapping was low. Demonstrate that contamination was not an issue.
Reply
The mapped reads must have been low because this is the closest available reference genome we have to our samples.
Comment
Further, in lines 430-432 and Table 4, the reported heterozygosity is 0.782, which is very high for a fungal pathogen. The pipeline used GATK HaplotypeCaller (default diploid assumptions), yet S. rolfsii is generally haploid/heterokaryotic. Treating isolates as diploid may inflate heterozygosity estimates. Please state expected ploidy of isolates, re-run or re-interpret SNP calling under correct assumptions, and provide allele frequency/allele balance plots to exclude artifacts or mixed cultures.
Reply
Indeed, thanks for noticing this. During variant calling, we realized the ploidy of our samples was set to 2. We corrected this. We have re-conducted the variant calling with the right parameter sets and the diversity outputs are now okay
Comment
AMOVA shows 87.6% of variance “within samples” and only 0.39% between populations, yet the manuscript describes 5 genetic clusters as distinct. The high within-sample variance likely reflects heterokaryosis or analysis artifacts. Please clarify biological meaning of “within sample” variance. Reconcile AMOVA with clustering results. Avoid over-interpretation of weak structure.
Reply
Thanks for noticing the above. The oversight during variant calling also affected the AMOVA results. From the plot of Bayesian information content, we can see that there are 4 to 5 genetic clusters. We re-conducted AMOVA using the new VCF, and the variation between clusters is now 53.9 while that within clusters was 46.1. We have replaced “between sample” with between population/Cluster and “within samples” with within population/cluster
Comment
Other than these concerns, the authors have used too many abbreviations, I would strongly recommended to use the complete name at first mention and later on, use only abbreviation. Importantly, standardize agro-ecological zone abbreviations and ensure they match in all tables/figures (EH, LVC, NMFS, SWH, TFZ, WMFS, WNMFS).
Reply
We have fixed the above concern
Comment
There are too many figures and tables, I would suggest shifting some figures and tables to supplementary data, figures up to 6 and tables up to 3-4/5 would be sufficient.
Reply
We have moved some tables and figures to supplemental materials
Comment
In the discussion section, only few references were cited, a strong discussion provides the support so, I would suggest expanding discussion based on the results of the present study.
Reply
The discussions have been expanded

Reviewer 2 Report
Article: Pathogenic and Genetic Diversity of Sclerotium rolfsii, the Causal Agent of Southern Blight of Common Bean in Uganda
In this study, the authors assessed the genetic diversity of 188 Sclerotium rolfsii strains using single-nucleotide polymorphisms obtained through whole-genome sequencing. Furthermore, the morphology and pathogenicity of the studied strains were characterized. Overall, the study was well planned, but testing the plant's virulence against different strains would have improved the work. The results are well presented and described. However, the text contains typos and errors. I have highlighted the errors in the attached file, as they are not worth describing separately. Below is a list of questions and comments.
Figure 1. What does the arrow mean?
Lines 193, 197, 200, 206. Specify G, not rpm.
Line 196. Specify the manufacturer: Geno grinder.
Line 218. Specify manufacturer: NovaSew 6000.
Lines 273-284. Provide a link to a table with this information.
Figures 3, 8. Remove unnecessary letter designations from the figures.
Figures 4, 8. Describe the statistical analysis in the figure caption.
Figure 5. Split this figure into several separate ones.
Figure 5 C(A). Why is K132 used instead of Cal96 in this case?
Figure 6C. I suggest presenting this data in a table or providing a high-resolution figure in the supplementary materials. The cluster composition is not clear from the figure provided.
Line 495. Table S5 is missing.

Author Response
The manuscript has been attached
Reviewer 2 comments
In this study, the authors assessed the genetic diversity of 188 Sclerotium rolfsii strains using single-nucleotide polymorphisms obtained through whole-genome sequencing. Furthermore, the morphology and pathogenicity of the studied strains were characterized. Overall, the study was well planned, but testing the plant's virulence against different strains would have improved the work. The results are well presented and described. However, the text contains typos and errors. I have highlighted the errors in the attached file, as they are not worth describing separately. Below is a list of questions and comments.
Figure 1. What does the arrow mean?
Reply: The arrow was there accidentally
Lines 193, 197, 200, 206. Specify G, not rpm.
Line 196. Specify the manufacturer: Geno grinder.
Reply: Manufacturer specified
Line 218. Specify manufacturer: NovaSeq 6000.
Reply: Manufacturer specified
Lines 273-284. Provide a link to a table with this information.
Reply: Link provided (Table S2)
Figures 3, 8. Remove unnecessary letter designations from the figures.
Reply: Unnecessary letters removed
Figures 4, 8. Describe the statistical analysis in the figure caption.
Reply: Statistical methods used added
Figure 5. Split this figure into several separate ones.
Reply: If I split figure five, the list of figures will become too many. The figure has been replaced with others
Figure 5 C(A). Why is K132 used instead of Cal96 in this case?
Reply: It's another name for CAL96 which I had specified in the materials and methods. I have now instead used a table to present the data
Figure 6C. I suggest presenting this data in a table or providing a high-resolution figure in the supplementary materials. The cluster composition is not clear from the figure provided.
Reply: It is impossible to present a high-resolution figure since the isolates are too many. I have presented the table with genetic clusters of isolates in Table S4
Line 495. Table S5 is missing.
Reply: Table S5 is not supposed to be there. However, there is a new table S5 after addressing the reviewers' comments

Round 2
Reviewer 1 Report
Thank you for sharing the revised version of this interesting manuscript. The authors have satisfactorily addressed most of the previous concerns, and the overall quality of the paper has improved considerably. The study presents clear novelty and uniqueness in its approach and findings. I have no further comment.
I have no further comment.